# MultiModalPFN: Extending Prior-Data Fitted Networks for Multimodal Tabular Learning

## Abstract

Recently, TabPFN has gained attention as a foundation model for tabular data. However, it struggles to integrate heterogeneous modalities such as images and text, which are common in domains like healthcare and marketing, thereby limiting its applicability. To address this, we present the Multi-Modal Prior-data Fitted Network (MMPFN), which extends TabPFN to handle tabular and non-tabular modalities in a unified manner. MMPFN comprises per-modality encoders, modality projectors, and pre-trained foundation models. The modality projectors serve as the critical bridge, transforming non-tabular embeddings into tabular-compatible tokens for unified processing. To this end, we introduce a multi-head gated MLP and a cross-attention sampler that extract richer context from non-tabular inputs while mitigates attention imbalance issue in multimodal learning. Extensive experiments on medical and general-purpose multimodal datasets demonstrate that MMPFN consistently outperforms competitive state-of-the-art methods and effectively exploits non-tabular modalities alongside tabular features. These results highlight the promise of extending prior-data fitted networks to the multimodal setting, offering a scalable and effective framework for heterogeneous data learning.

## 1 Introduction

Tabular data is one of the most widely used data types in domains such as healthcare and marketing. Traditionally, tree-boosting algorithms (Dorogush et al., 2018) have dominated this field due to their fast training and strong predictive performance. However, with the advent of TabPFN (Hollmann et al., 2022) and various tabular deep learning models (Somepalli et al., 2021; Bahri et al., 2021), it has become clear that deep learning can achieve superior results in learning tabular representations. This progress has expanded tabular data analysis into a broader range of applications, where it is often combined with unstructured data (Hager et al., 2023). For instance, diagnostic tasks may jointly leverage structured test results and medical images (Huang et al., 2020; Schilcher et al., 2024), while marketing tasks may combine numerical sales records with textual product reviews (Das et al., 2024; Sukel et al., 2024). Despite this growing interest, attempts to extend tree-boosting algorithms such as CatBoost to heterogeneous data types have yielded only modest gains. In parallel, deep learning models that embed and jointly process tabular data with images or text have shown potential, but they often suffer from limited performance and slow training (Cui et al., 2023; Zhao et al., 2024). A representative example, TabPFN, excels on purely tabular tasks but remains limited in its ability to directly integrate unstructured modalities.

We propose the **Multi-Modal Prior-data Fitted Network (MMPFN)**, an extension of TabPFN that processes tabular and non-tabular modalities in a unified manner. MMPFN first extracts features with per-modality encoders for tabular, image, and text inputs: the original TabPFN encoder for tabular data and modality-specific pretrained foundation models for non-tabular data. MMPFN then align these embeddings using a novel modality projector, which maps non-tabular embeddings into the tabular embedding space. The projected non-tabular embeddings are concatenated with tabular embeddings and jointly processed by the pre-trained TabPFN backbone. This plug-and-play design enables seamless integration of diverse foundation models and improves performance via fine-tuning, yielding faster convergence and more stable training.

In addition, we address two failure modes in multimodal learners: (i) overcompressed non-tabular embeddings (e.g., a single $[CLS]$ token) and (ii) attention imbalance induced by token-count disparities across modalities. We theoretically and empirically show that MMPFN is also susceptible to the latter. To mitigate overcompression, we introduce a multi-head gated MLP (MGM) that expands and enriches non-tabular representations into multiple tokens. We then develop a cross-attention sampler (CAS) that merges a compact, informative subset, thereby rebalancing attention. MGM and CAS build our modality projector.

We evaluate MMPFN on medical and general-purpose benchmarks that pair tabular inputs with images or text inputs. Across nearly all datasets, MMPFN surpasses recent state-of-the-art multimodal methods (Hager et al., 2023; Du et al., 2024; Hemker et al., 2024; Luo et al., 2025; Bonnier, 2024) and strong AutoML baselines (Tang et al., 2024b). Extensive experiments demonstrate that MGM and CAS effectively mitigate the identified failure modes. In addition, we confirm that MMPFN scales positively as modalities are added and preserves the strengths of TabPFN's attention-based modeling, delivering robust performance in low-data regimes. Furthermore, MMPFN reduces the training time by an order of magnitude.

Our main contributions are summarized as follows:

- We propose MMPFN, the first framework to extend TabPFN to heterogeneous inputs (tabular + image/text) through a unified pathway.

- We identify two failure modes in multimodal learners: overcompressed non-tabular embeddings and token-count–induced attention imbalance. To overcome this, we introduce MGM and CAS as components of the modality projector.

- Through extensive experiments across medical and general-purpose datasets, we show that MMPFN consistently outperforms competitive baselines, scales positively as modalities are added, and maintains robust performance under data scarcity and limited compute.

## 2 RELATED WORK

**Vision–Language Multimodal Models**  Early research in multimodal learning developed fusion and conditioning mechanisms for integrating text and images. FiLM (Perez et al., 2018) introduced feature-wise modulation for language-conditioned visual reasoning, while early transformer-based models such as ViLBERT, VisualBERT, VL-BERT, LXMERT, and UNITER (Lu et al., 2019; Li et al., 2019; Su et al., 2019; Tan & Bansal, 2019; Chen et al., 2020) explored co-attention and unified architectures, achieving state-of-the-art results on vision–language benchmarks. A major shift came with CLIP (Radford et al., 2021), which used large-scale contrastive pretraining for scalable zero-shot transfer. More recent approaches, such as BLIP-2 (Li et al., 2023) and LLaVA (Liu et al., 2023), integrated large language models for generalizable multimodal reasoning.

**Tabular and Multimodal Models**  The pretraining-driven paradigm has since expanded to structured data. In the tabular domain, approaches typically adopt either a *row-as-text* strategy, serializing entire rows for large language model (LLM) processing (Hegselmann et al., 2023), or a *per-column embedding* strategy with modality-specific encoders. Methods such as Tab2Text (Lin et al., 2024) transform rows into textual narratives for improved alignment, while others (Bonnier, 2024) demonstrate that careful design of fusion layers substantially improves benchmarks. LANISTR (Ebrahimi et al., 2023) extended this direction with similarity-based multimodal masking, enabling joint learning from language, images, and structured inputs even with missing modalities.

Unstructured–structured integration has also been explored in image-centric datasets. Representative works include MMCL (Hager et al., 2023), which aligned tabular and image embeddings with contrastive learning; TIP (Du et al., 2024), which improved robustness to incomplete features; STiL (Du et al., 2025), which leveraged unlabeled data via semi-supervised pseudo-labeling; TIME (Luo et al., 2025), which employed TabPFN (Hollmann et al., 2022) as a tabular encoder; and Turbo (Jiang et al., 2025), which strengthened cross-modal reasoning. Beyond individual models, toolkits such as AutoGluon (Tang et al., 2024a) and modular pipelines (Gu & Budhkar, 2021) provide practical infrastructure for integrating text, image, and tabular features.

Despite progress, most multimodal approaches remain concentrated on vision–language tasks, with systematic treatment of structured data still limited. Fusion strategies are often heuristic, with weak guarantees under low-data regimes or modality imbalance. Addressing these gaps is critical for building generalizable multimodal tabular systems.

**General-Purpose Pre-trained Models** Pretraining large foundation models has transformed representation learning across domains. In NLP, models progressed from masked language modeling to more efficient self-supervised strategies such as ELECTRA (Clark et al., 2020) and DeBERTa (He et al., 2021b). Later refinements including DeBERTaV3 (He et al., 2021a), ModernBERT (Warner et al., 2024), and multilingual encoders such as BGE/M3 (Chen et al., 2023) introduced architectural improvements (e.g., disentangled embeddings, FlashAttention-2, optimized tokenization) and broadened applications to retrieval and cross-lingual tasks.

In computer vision, self-supervised pretraining emerged as the dominant paradigm. DINOv2 (Oquab et al., 2023) and DINOv3 (Siméoni et al., 2025) demonstrated scalable self-distillation for robust visual features, EVA (Fang et al., 2022; 2024) advanced masked image modeling with large Vision Transformers, and iBOT (Zhou et al., 2022) combined masking with self-distillation for effective ViT representation learning.

For structured data, TabPFN (Hollmann et al., 2022; 2025) extended this paradigm by pretraining on large synthetic datasets, learning a general prior over tabular distributions. This enables strong performance on small- and medium-sized datasets in a single forward pass without task-specific fine-tuning, positioning TabPFN as a foundation model for tabular learning.

Taken together, these advances demonstrate the effectiveness of pretraining. Yet, compared to NLP and vision, pretraining for multimodal tabular data remains underexplored. Bridging this gap—particularly for multimodal integration with structured inputs—is a key step toward more comprehensive foundation models.

## 3 PRELIMINARY

TabPFN (Hollmann et al., 2022; 2025) is a tabular foundation model that treats tabular learning as amortized Bayesian inference: a transformer pretrained on synthetic tabular datasets sampled from structural causal model priors, which jointly processes a training set and query set to produce posterior-predictive label distributions in one forward pass.

Architecturally, TabPFN stacks 2D TabPFN blocks. Each block splits attention into two steps: feature attention, where a feature looks at other features in the same sample; and sample attention, where the same feature looks across all samples. This design is permutation-invariant over samples and features and scales efficiently to larger tables than those seen during training. An MLP follows the attention steps. All sublayers use residual connections and layer normalization.

For in-context inference, TabPFN processes the concatenated training and test rows with masks that allow self-attention within labeled training rows and restrict test rows to cross-attend only to training rows. A lightweight MLP head then maps the resulting test embeddings to predictions.

## 4 MULTIMODAL PFN

### 4.1 ARCHITECTURE

We extend TabPFN to the multimodal setting, where image or text modalities accompany tabular inputs. Section 3 outlines our multimodal PFN (MMPFN), which consists of per-modality encoders, a modality projector, a TabPFN backbone. The encoders map each modality to a feature vector. The modality projector aligns image and text features in a shared embedding space. TabPFN then processes the resulting multimodal embeddings, and the decoder produces predictions for the test samples.

**Per-Modality Encoders** The per-modality encoders comprise tabular, image, and text branches. The tabular branch is identical to the TabPFN v2 encoder and remains frozen during fine-tuning. For images, we use the DINOv2 ViT-B/14 backbone. Thus, the inputs are resized so that height

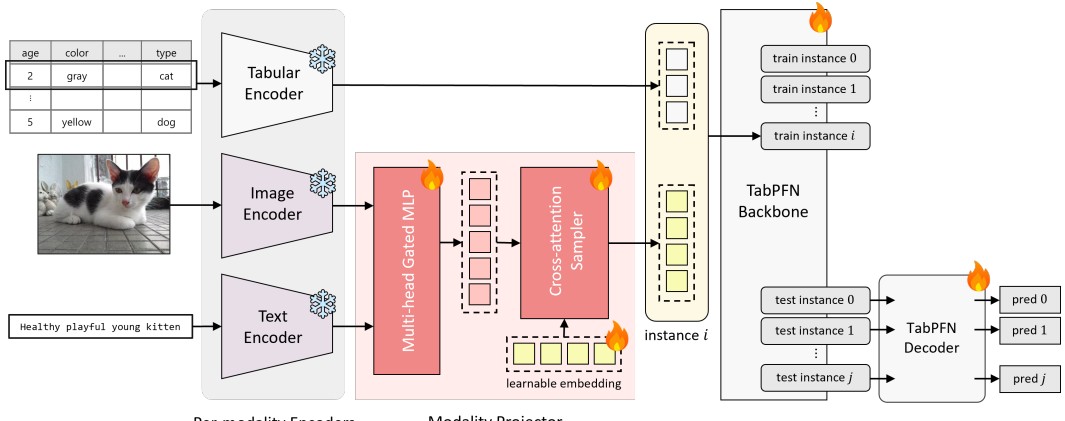

Figure 1: The architecture of MMPFN. MMPFN extends TabPFN by incorporating per-modality encoders and a modality projector to extract features from non-tabular data. Newly developed components are highlighted in color, while existing ones appear in gray. Layers marked as 'frozen' remain fixed during fine-tuning, whereas all others are trainable. Encoded target labels are part of the training inputs but are omitted from the diagram for clarity.

and width are divisible by 14, and we take the `[CLS]` embedding as a global representation. For text, we adopt ELECTRA based on empirical comparisons (e.g., outperforming DeBERTa). Text is tokenized with sequences truncated to 512 tokens, and the `[CLS]` embedding is used as the text representation.

**Modality Projector** The modality projector transforms image and text embeddings to tabular-like representation, which share $d$-dimensional space compatible with the TabPFN backbone. As shown in Section 3, it comprises two sublayers: a multi-head gated MLP (MGM) and a cross-attention sampler (CAS). The MGM addresses the limitation of a single `[CLS]` embedding, which can overly compress image/text information, by expanding it into $N$ parallel $d$-dimensional projections. Specifically, the `[CLS]` embedding is fed into $N$ lightweight MLP heads, and a Gated Linear Unit (GLU) (Dauphin et al., 2017) modulates each head's contribution, preserving diverse features in the compact representations. The resulting set of projected tokens are then passed to the CAS.

The CAS balances tabular and non-tabular cues before fusion in the backbone. It applies cross-attention with $K$ learnable queries to summarize the non-tabular embeddings (image/text) into $K$ representative $d$-dimensional embeddings. An excessive number of non-tabular tokens can degrade performance due to attention imbalance, where a modality with many tokens dominates the computation. CAS mitigates this by producing a compact, calibrated set of embeddings for the TabPFN backbone. Section 4.3 will discuss the attention imbalance issue in MMPFN.

## 4.2 TRAINING

Since TabPFN is pre-trained on large corpora of synthetic tabular data, its representations can be misaligned with image/text embeddings. We therefore freeze all modality encoders and train the modality projector, the TabPFN backbone, and the decoder. Note that all components are pre-trained, except for the modality projector. To leverage TabPFN's in-context inference, we follow its standard protocol: split the multimodal data into training and test sets, concatenate their embeddings into a single table, and feed it to the backbone. The model then produces predictions for the test samples to obtain supervisory signals for training.

Table 1: Comparison with state-of-the-arts on image–tabular multimodal datasets. Performance is reported as "accuracy (rank)", where the rank represents the ordering of the methods according to their accuracy within each dataset (lower rank indicates better performance). Results are averaged over five random seeds. Best accuracy is shown in **bold**, second best in underline. "MASS" and "Calc" denote the respective CBIS-DDSM tasks.

| Method | Modality | PAD–UFES–20 | MASS | Calc | Petfinder | Avg. rank |
|---|---|---|---|---|---|---|
| TabPFN (Hollmann et al., 2022) | T | 82.17 (2) | 71.27 (5) | 73.31 (2) | 36.33 (8) | 4.25±2.87 |
| Catboost (Dorogush et al., 2018) | T+I | 80.43 (4) | **78.31 (1)** | 72.09 (4) | 38.69 (4) | 3.25±1.50 |
| AutoGluon (Tang et al., 2024a) | T+I | 81.09 (3) | 76.28 (2) | 71.04 (6) | 38.81 (3) | 3.50±1.73 |
| MMCL (Hager et al., 2023) | T+I | 76.61 (7) | 57.62 (7) | 60.12 (8) | 36.61 (7) | 7.25±0.5 |
| TIP (Du et al., 2024) | T+I | 78.75 (6) | 73.12 (4) | 67.96 (7) | 37.28 (5) | 5.50±1.29 |
| HEALNet (Hemker et al., 2024) | T+I | 74.65 (8) | 68.10 (6) | 71.83 (5) | 37.03 (6) | 6.25±1.26 |
| TIME (Luo et al., 2025) | T+I | 80.35 (5) | - | 72.70 (3)[1] | 39.25 (2) | 3.33±1.53 |
| **MMPFN (Ours)** | T+I | **84.87 (1)** | 75.10 (3) | **76.07 (1)** | **40.74 (1)** | **1.50±1.00** |

## 4.3 Attention Imbalance in MMPFN

We study how the number of non-tabular tokens affects attention mechanism. Consider a query token $q$ attending to two sets of keys: non-tabular tokens $k_1^{(I)}, \cdots, k_{N_I}^{(I)}$ and tabular tokens $k_1^{(T)}, \cdots, k_{N_T}^{(T)}$, where $N_I$ and $N_T$ are their respective counts. The scaled dot-product attention scores are given by

$$s_i^{(I)} = q^\top k_i^{(I)}/\sqrt{d}, \qquad s_j^{(T)} = q^\top k_j^{(T)}/\sqrt{d}. \tag{1}$$

Let $w_i^{(I)} = e^{s_i^{(I)}}$ and $w_j^{(T)} = e^{s_j^{(T)}}$ be the unnormalized attention weights, and define the per-token expectations $c_I = \mathbb{E}[w_i^{(I)}]$ and $c_T = \mathbb{E}[w_j^{(T)}]$, where the expectation is over token indices and any randomness in $(q, k)$. Also, let $a_I$ denote the total attention weight allocated to the non-tabular set, defined by

$$a_I = \sum_{i=1}^{N_I} \frac{w_i^{(I)}}{\sum_{u=1}^{N_I} w_u^{(I)} + \sum_{v=1}^{N_T} w_v^{(T)}}. \tag{2}$$

Then its expectation is approximated by

$$\mathbb{E}[a_I] \approx \frac{N_I c_I}{N_I c_I + N_T c_T} \tag{3}$$

Hence, when per-token quality is comparable ($c_I \approx c_T$), token-count imbalance ($N_I > N_T$) induces attention imbalance, potentially degrading performance. Consequently, MMPFN's performance might vary with the modality token ratio. This suggests the importance of CAS. In Section 5, we empirically validate this observation by varying $K$ in the CAS.

## 5 Experiments

### 5.1 Experimental Setup

We evaluated MMPFN on classification tasks across multiple multimodal datasets combining either image–tabular or text–tabular features, and compared its performance with prior models. Our goal was twofold: (i) to verify that MMPFN achieves superior performance in multimodal input settings compared to existing methods, and (ii) to examine whether the inclusion of unstructured inputs leads to performance gains over the tabular-only baseline. To this end, we also compared MMPFN with fine-tuned TabPFN models trained on the same datasets. MMPFN either surpasses strong baselines or attains performance on par with them depending on dataset characteristics, thereby ensuring consistent competitiveness on image–tabular datasets, and on text–tabular datasets. More Detailed experimental settings are provided in the Appendix A.1

Table 2: Comparison with state-of-the-arts on text–tabular multimodal datasets. Performance is reported as "accuracy (rank)", where the rank represents the ordering of the methods according to their accuracy within each dataset (lower rank indicates better performance). Results are averaged over five random seeds. Best accuracy is shown in **bold**, second best in underline. The results of AllTextBERT, TFN, MulT, and TTT were cited from Bonnier (2024).

| Method | Modality | Airbnb | Salary | Cloth | Petfinder | Avg. rank |
|---|---|---|---|---|---|---|
| TabPFN (Hollmann et al., 2022) | T | 46.96 (2) | 44.96 (6) | 55.07 (8) | 36.33 (6) | 5.5±2.52 |
| Catboost (Dorogush et al., 2018) | T+t | 43.56 (4) | 40.36 (8) | 59.24 (7) | 35.47 (7) | 6.5±1.73 |
| AutoGluon (Tang et al., 2024a) | T+t | 44.60 (3) | 45.24 (5) | **72.07(1)** | 37.96 (3) | 3.0±1.63 |
| AllTextBERT (Bonnier, 2024) | T+t | 30.9 (8) | 44.0 (7) | 68.0(2) | 34.6 (8) | 6.25±2.87 |
| TFN (Zadeh et al., 2017) | T+t | 35.7 (7) | 45.8 (3) | 60.1 (6) | 36.8 (5) | 5.25±1.71 |
| MulT (Tsai et al., 2019) | T+t | 36.3 (6) | 45.4 (4) | 63.6 (5) | 37.6 (4) | 4.75±0.96 |
| TTT (Bonnier, 2024) | T+t | 38.3 (5) | **47.2 (1)** | 65.5 (4) | 38.9 (2) | 3.0±1.83 |
| **MMPFN (Ours)** | T+t | **47.78 (1)** | 45.87 (2) | 66.26 (3) | **39.04 (1)** | **1.75±0.96** |

## 5.2 MAIN RESULTS

**Results on Tabular–Image Modality Datasets** Table 1 summarizes the classification accuracy of MMPFN and state-of-the-art baselines on four tabular–image datasets. Compared with fine-tuned TabPFN (Hollmann et al., 2025), which use only tabular inputs, MMPFN consistently improves performance by leveraging image features. In Table 1, MMCL (Hager et al., 2023), TIP (Du et al., 2024), and HEALNet (Hemker et al., 2024) show inconsistent performance, due to small dataset size and the low dimensionality of tabular features. In contrast, MMPFN attains the best results on all datasets except MASS, where it is third-best. Compared with TIME (Luo et al., 2025), which uses TabPFN as its tabular encoder, MMPFN delivers substantial gains, indicating that our modality projection strategies are more effective than simple fusion. AutoGluon (Tang et al., 2024a), an AutoML framework for multimodal data, is competitive on several tabular benchmarks but consistently underperforms MMPFN across all evaluated datasets. CatBoost (Dorogush et al., 2018) used image embeddings as raw input features and achieved strong performance on MASS, but fell short of MMPFN on other datasets.

**Results on Tabular–Text Modality Datasets** Table 2 reports results on tabular–text modality datasets. As in the image setting, adding text features consistently outperforms the fine-tuned TabPFN baseline. MMPFN is particularly strong on *Airbnb*. This dataset includes 50+ tabular features and a single text field, allowing tabular-specialized models to capture most of the predictive signal. Accordingly, MMPFN substantially outperforms language model–based methods, such as TFN (Zadeh et al., 2017) and MulT (Tsai et al., 2019), which struggle to exploit the abundant tabular features. By contrast, in *Cloth*, the tabular part has few, weakly informative features, while the review text carries most of the signal. This is reflected in the poor performance of tabular-only models and the strong results of AllTextBERT (Bonnier, 2024), indicating that text-specialized models excel in such cases. Even so, among methods that explicitly preserve tabular structure, MMPFN achieves the best overall performance, trailing the text-specialized baseline by only a small margin. This contrasts with prior multimodal tabular studies, which emphasized tabular-dominant datasets(Hager et al., 2023). Overall, MMPFN is not merely strong on tabular inputs. It also models unstructured modalities effectively, establishing it as a truly multimodal approach.

---

[1] We cite all results of Luo et al. (2025) directly. Although TIME used the CBIS-DDSM dataset without specifying subtype, the reported sample size matches the calcification subset, so we list it under CBIS-DDSM calcification in Table 1. Since the code is unavailable, reproduction was infeasible.

[1] AllTextBert converts all tabular features into strings, concatenates them, and inputs the resulting sequence into `DistilBERT-base-uncased` for modeling, as described in (Bonnier, 2024).

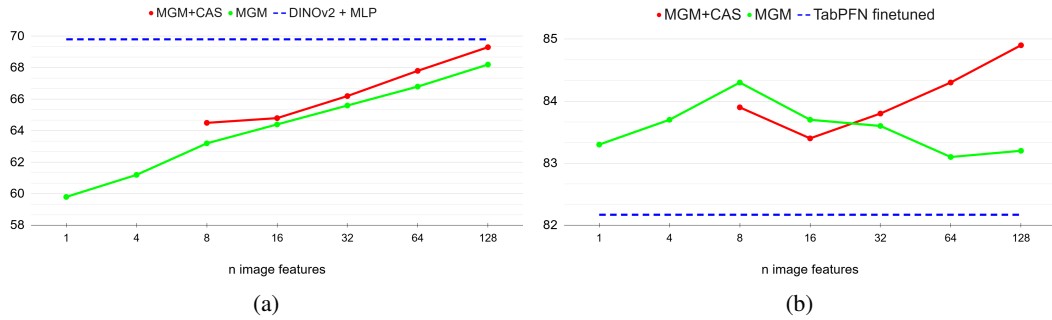

(a)  (b)

Figure 2: Performance on PAD–UFES–20 with varying image feature count and fixed tabular inputs. (a) Image-only experiments with DINOv2 + MLP as the baseline. (b) Image–tabular experiments with fine-tuned TabPFN as the baseline. In both settings, CAS compress $n$ image features into 8; the x-axis denotes the number of image features and the y-axis indicates accuracy.

Table 3: Ablation study: Accuracy across datasets under different feature extraction mechanisms. Best accuracy is shown in **bold**.

|  | Single-head | MGM w/o gating | Mixture-of-Experts | MGM |
|---|---|---|---|---|
| Calc | 73.48 | 73.10 | 71.35 | **76.07** |
| Cloth | 58.28 | 62.37 | 60.49 | **66.26** |

### 5.3  MMPFN as an Image Classification Model

Figure 2a evaluates MMPFN with *non-tabular–only* inputs. Specifically, we compare MMPFN against DINOv2 (Oquab et al., 2023), a popular image foundation model, using images from *PAD–UFES–20*. For the baseline, we extract the pre-trained DINOv2 $[CLS]$ embedding and attach an MLP head for classification—an approach that is widely adopted and often near-optimal (Sun et al., 2019).

With image embeddings as the sole inputs, MMPFN is within ∼1% of DINOv2 (69.30% vs. 69.89%). Accuracy increases consistently as the number of image feature tokens grows, indicating that additional tokens capture complementary, higher-resolution information. Thus, despite being trained with a tabular synthetic data, MMPFN functions effectively as a classifier over image embeddings cast into tabular-like features, highlighting seamless integration of non-tabular modalities with tabular data. We also note that the presence of CAS yields a modest additional gain for MMPFN.

### 5.4  Analysis

**Attention Imbalance**  We empirically study the attention imbalance in multimodal processing. Figure 2b reports MMPFN accuracy on PAD–UFES–20 (image+tabular) as the number of image features varies, comparing MGM (no CAS) and MGM+CAS. For reference, Figure 2b also includes the accuracy of the fine-tuned TabPFN (tabular-only) as a unimodal baseline.

In Figure 2b, we see that naively increasing the number of image tokens can hurt performance. MGM outperforms the TabPFN baseline by exploiting image context, but its accuracy peaks around $n=8$ image tokens and declines as $n$ grows. This contrasts with Figure 2a, where MGM operates in the image-only setting and does not exhibit the same drop. This non-monotonic trend is consistent with attention imbalance: when more image tokens are introduced, noise receives nonzero attention and the modality with more tokens captures a disproportionate share of the attention budget, diluting useful tabular signal.

The proposed CAS mitigates this effect by consolidating many image embeddings into a compact set (8 tokens in Figure 2b), preserving salient information while capping token count. As a result, MGM+CAS improves steadily with larger candidate sets and outperforms MGM across $n$, confirming that merging before attention yields more informative, balanced cross-modal representations.

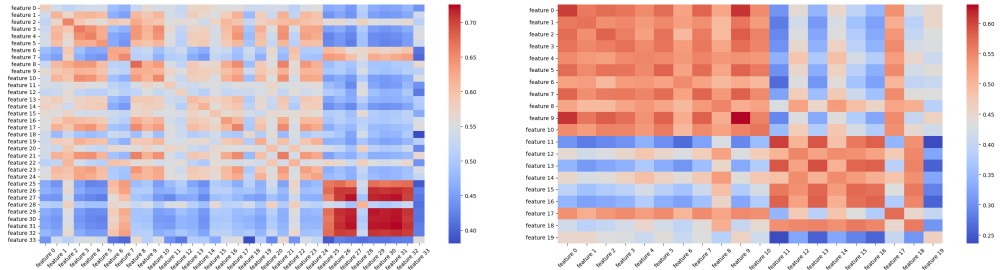

Figure 3: Cosine similarity between multimodal feature embeddings. Left: tabular vs. text features in the Airbnb dataset. Right: tabular vs. image features in the PAD–UFES–20 dataset. Axes denote all tabular and text/image features

Table 4: Performance in Low-data Regime. Reported as mean accuracy over five random seeds, with parentheses showing percentage change relative to full-data training. The 'average' column denotes the mean of all percentage changes. Best accuracy is shown in **bold**.

|  | PAD–UFES–20 | MASS | Calc | PetFinder(image) | average |
|---|---|---|---|---|---|
| TIP 10% | 70.44(-10.6) | 68.31(-6.58) | 62.27(-8.37) | 34.86(-6.49) | -8.00 |
| MMPFN 10% | **72.87**(-14.14) | **76.13**(+0.75) | **72.09**(-5.23) | **35.73**(-12.30) | -10.27 |

**Modality Projector**   Table 3 compares Calc and Cloth accuracy across projector variants. (1) Single-head: a single linear layer projects the non-tabular $[CLS]$ token. (2) MGM w/o gating: GLU in MGM is replaced with GELU, removing the multiplicative linear gate. (3) Mixture-of-Experts (MoE): a learned gate routes inputs so only a subset of experts is active per example. Experimental details are in Appendix A.4.

First, MMPFN substantially outperforms the single-head baseline, highlighting the benefit of multi-head projection. Although the single-head variant surpasses fine-tuned TabPFN (tabular-only) in Table 1, indicating complementary non-tabular signal, it relies on an overcompressed $[CLS]$ representation leads to suboptimal multimodal learning. Second, MMPFN consistently exceeds MGM w/o gating baseline, suggesting that GLU's element-wise gating increases feature diversity and strengthens representation learning. Appendix A.5 will provide detailed discussion about it. Finally, MGM significantly excels MoE across datasets, validating our projector design that first extracts multiple embeddings and then selectively merges them via CAS.

**Correlation Analysis of Cross-Modal Embeddings**   Figure 3 visualizes cosine similarities among TabPFN–backbone embeddings on Airbnb (tabular–text) and PAD–UFES–20 (tabular–image). These similarities illustrates the predictive relationships between features learned by MMPFN. As expected, within-modality blocks exhibit high similarity. However, several tabular–image/text pairs are also strongly aligned, indicating that MMPFN models cross-modal interactions rather than only within-modality structure. Details of the cosine similarity computation are provided in Appendix A.6.

**Robustness in Low-Data Regimes**   Tabular datasets often require expert annotation, resulting in limited sample sizes and sparse labels (Du et al., 2024; 2025). Consequently, models that remain robust under data scarcity are highly desirable. In such low-data regimes, MMPFN demonstrates strong performance. Table 4 compares the results of MMPFN and TIP when trained on only 10% of randomly selected samples from each dataset. While TIP employs self-supervised pretraining using all unlabeled data, our evaluation focuses on supervised finetuning under limited labeled data.

The analysis shows that, although the relative performance drop was larger for MMPFN, it consistently outperformed TIP across all datasets, even when trained with only 10% of the data. Interestingly, on the CBIS-DDSM MASS dataset, performance improved under subsampling. This behavior suggests that the PFN, pretrained on synthetic priors, can better capture discriminative characteris-

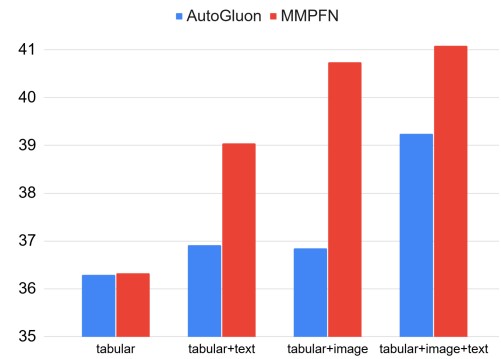 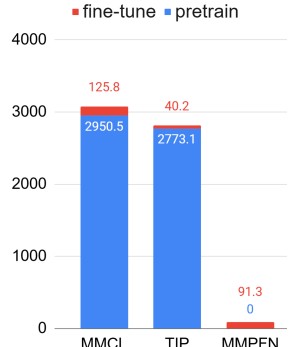

Figure 4: Accuracy of AutoGluon vs. MMPFN on PetFinder under different modality combinations (tabular, +text, +image, +image+text).

Figure 5: Training time (sec.) on Calc. For MMCL and TIP, pretraining and fine-tuning times are reported separately.

tics when finetuned on a smaller set of labeled examples. Additional details on the behavior of MMPFN in low-data settings are provided in Appendix A.7.

**Scaling with Added Modalities**  We assess MMPFN as multiple non-tabular modalities are added. On Petfinder, we compare against AutoGluon, a multimodal AutoML system supporting image and text modalities. As shown in Figure 4, MMPFN's accuracy increases monotonically from *tabular → tabular+text → tabular+image → tabular+image+text* (39% → 40% → 41%), indicating complementary signal from both image and text. These results have particular significance for tabular modeling, where performance improvements from architectural changes alone are often saturated. Adding complementary modalities offers a practical route to further gains. Moreover, MMPFN outperforms AutoGluon under every combination. Unlike AutoGluon's large ensembles, MMPFN achieves higher accuracy with a lightweight and specialized architecture.

**Efficiency**  Next, we evaluate the training efficiency of MMPFN by comparing training time with exiting methods: MMCL and TIP. Although these models are built with the goal of performing well under limited training data, their contrastive learning strategies still depend on large pre-training corpora and lengthy optimization schedules. MMPFN avoids this overhead by reusing a pretrained tabular foundation backbone (TabPFN) and fine-tuning only the multimodal layers (MGM, CAS) with the backbone—no extra pretraining.

As shown in Figure 5, MMPFN delivers large speedups on Calc dataset while maintaining strong accuracy: it trains in 91.3 s with zero pretraining, versus MMCL at 2950.5 + 125.8 s and TIP at 2773.1 + 40.2 s—i.e., 3.0% and 3.2% of their total time. These gains stem from using TabPFN as a tabular foundation model. However, very large datasets can stress TabPFN's scalability, and generating many MGM tokens or fine-tuning upstream image/text encoders would increase cost. Despite these limitations, MMPFN consistently achieves competitive or superior accuracy with substantially shorter training times than contrastive baselines.

## 6 CONCLUSION

We introduced MMPFN, a multimodal extension of TabPFN that unifies tabular, image, and text inputs with per-modality encoders, a modality projector, and the TabPFN backbone. We developed MGM and CAS, which map non-tabular embeddings to the tabular space and mitigate token-count–induced attention imbalance. By leveraging pretrained foundation models and fine-tuning lightweight components, MMPFN achieved strong accuracy with substantially lower training costs. Across medical and general-purpose benchmarks, it consistently outperformed competitive state-of-the-art methods, scaled positively as modalities were added, and maintained robust performance in low-data regimes.

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

# A APPENDIX

## A.1 IMPLEMENTATION DETAILS

Table 5: Implementation details

| dataset | features_per_group | mgm_heads | CAS_heads | task_type |
|---|---|---|---|---|
| PAD–UFES–20 | 2 | 128 | 12 | multiclass |
| cbis_ddsm(Mass) | 1 | 128 | 8 | binary |
| cbis_ddsm(Calc) | 2 | 16 | 8 | binary |
| Airbnb | 2 | 128 | 24 | multiclass |
| Salary | 1 | 64 | 4 | multiclass |
| Cloth | 2 | 64 | 32 | multiclass |
| Petfinder Adoption (T+I) | 1 | 128 | 16 | multiclass |
| Petfinder Adoption (T+t) | 2 | 32 | 8 | multiclass |
| Petfinder Adoption (T+I+t) | 1 | 64 | 16 | multiclass |

All experiments were fine-tuned with the following default settings: learning rate of $1 \times 10^{-5}$, batch size of 1, maximum training steps of 100, and validation metric of log_loss. The mixer type was fixed to MGM+CAS, and random seeds $\{0, 1, 2, 3, 4\}$ were used throughout.

Task-specific parameters such as mgm_heads, CAS_heads, task_type, and features_per_group were adjusted as summarized in Table X. While features_per_group must be modified directly in the initialization of the PerFeatureTransformer, the remaining parameters can be provided as arguments to the fine_tune_mmpfn function.

For dataset handling, categorical feature indices were not explicitly set in the MMPFN configuration. Since specifying them generally improves performance, this should be enabled in practical applications. For datasets without predefined splits, such as CBIS–DDSM, 20% of the training data was reserved as a test set. In the DINOv2 baseline with an MLP classification head, data were split 8:2 into training and test sets, with 10% of the training set further used for validation.

No specialized weight initialization methods (e.g., Xavier) were applied, though such techniques are expected to improve performance. For comparative models based on contrastive pretraining (Hager et al., 2023; Du et al., 2024; 2025), training was conducted for 500 epochs using a cosine annealing scheduler with a 10-epoch warmup. In contrast, MMPFN was fine-tuned for only 100 steps with a fixed learning rate of $1 \times 10^{-5}$, employing ScheduleFree (Defazio et al., 2024) for adaptive learning rate scheduling.

**Text Data Preprocessing** All preprocessing procedures followed the implementations provided in the TTT codebase. For the Salary dataset, the original source URL referenced in Bonnier (2024) is no longer accessible; thus, we used a Kaggle-hosted copy. However, applying the official TTT preprocessing scripts to this version did not yield the dataset size reported in their paper, indicating possible discrepancies. Consequently, we re-evaluated the TTT baseline on this dataset using the new version. The resulting accuracy (46.5) was comparable to the originally reported performance, confirming that the revised dataset can be reliably used for evaluating our model.

Since both Electra and DeBERTa text encoders are restricted to 512 input tokens, sequences longer than this limit were truncated. For datasets containing multiple text attributes, we extracted embeddings for each attribute separately and incorporated them as additional text features. In contrast, TTT concatenates all text columns into a single sequence. Our approach yielded small but consistent accuracy improvements. However, since the gains were within the error margin, we did not include these results in the main comparison table.

The Airbnb and PetFinder datasets contain Chinese characters in their text columns. Because the ELECTRA encoder variant we used was not pretrained on Chinese, these tokens were replaced with empty strings prior to encoding. For CatBoost and AutoGluon, we employed the library's built-in text handling functionality.

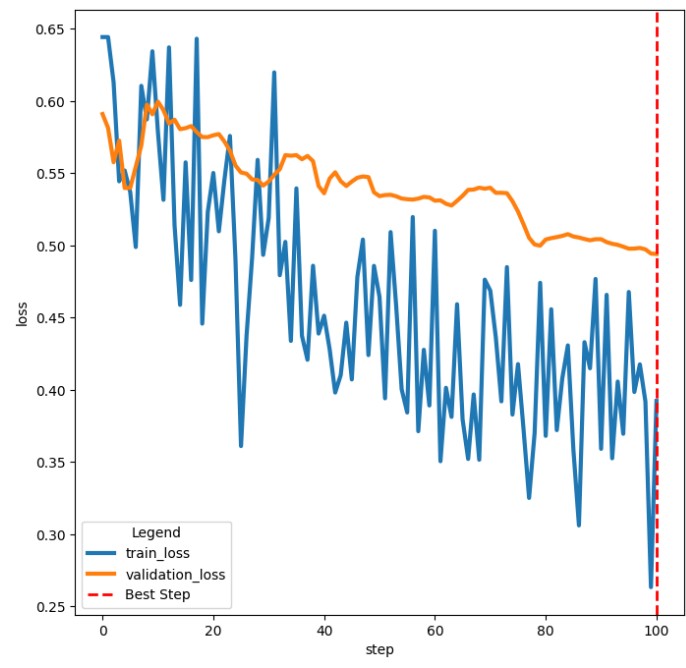

Figure 6: Fine-tuning loss curve of the MMPFN model trained on the PAD-UFES-20 dataset

## A.2 FINE-TUNING PROCEDURE AND RESULTS

Since TabPFN is pretrained with synthetic priors, it can be fine-tuned on downstream tasks using available training data. For fine-tuning, we adopted the official GitHub library[2] and modified it to support MMPFN. Specifically, we extended the `MMPFNClassifier` and `MMPFNRegressor` classes by introducing initialization parameters (`mgm_heads`, `CAS_heads`, and `mixer_type`) and enabling the models to accept multimodal embeddings as input.

Fine-tuning was conducted by splitting the training data into train and validation sets, and updating model parameters based on the validation loss. We employed a small learning rate ($1 \times 10^{-5}$) with a fixed maximum of 100 training steps. For learning rate scheduling, we used the `schedulefree` library. The fine-tuning loss curve for the PAD–UFES–20 dataset is illustrated in Figure 6.

## A.3 DATASETS

We employed a set of well-established benchmark datasets that have been extensively validated in prior research (Mráz et al., 2025; Tang et al., 2024b;a; Bonnier, 2024; Jiang et al., 2024). Leveraging such widely recognized datasets not only ensures reproducibility and comparability with existing methods but also provides a robust foundation for evaluating the effectiveness of our proposed approach under standardized experimental conditions. All datasets were randomly split into train and test sets, except for CBIS-DDSM, where the predefined train–test split was used.

**PAD–UFES–20** The PAD–UFES–20(Pacheco et al., 2020) dataset contains 2,298 samples of six skin lesion types, each paired with a clinical image and up to 26 metadata features such as age, lesion location, and diameter. Three lesion types correspond to cancers (Basal Cell Carcinoma, Squamous Cell Carcinoma including Bowen's disease, and Melanoma), while the remaining three are noncancerous (Actinic Keratosis, Nevus, and Seborrheic Keratosis), with approximately 58% of samples biopsy-proven. The images are provided in .png format and were collected using different smartphones, thereby resulting in varying image sizes that require preprocessing. https://data.mendeley.com/datasets/zr7vgbcyr2/1

---

[2]https://github.com/LennartPurucker/finetune_tabpfn_v2

Table 6: Dataset details

|  | PAD–UFES–20 | CBIS_DDSM(MASS) | CBIS_DDSM(Calc) | Petfinder Adoption | Airbnb | Salary | Cloth |
|---|---|---|---|---|---|---|---|
| n train samples | 1838 | 1318 | 1546 | 11721 | 5465 | 3837 | 18102 |
| n test samples | 460 | 378 | 326 | 2931 | 1367 | 960 | 4526 |
| n feats | 21 | 8 | 8 | 19 | 50 | 4 | 5 |
| n num feats | 3 | 3 | 3 | 5 | 27 | 1 | 2 |
| n cat feats | 18 | 5 | 5 | 14 | 23 | 3 | 3 |
| n images | 1 | 3 | 3 | 1 | 0 | 0 | 0 |
| n texts | 0 | 0 | 0 | 1 | 1 | 3 | 3 |
| n classes | 6 | 2 | 2 | 5 | 10 | 6 | 5 |

**CBIS–DDSM**    The CBIS–DDSM(Sawyer-Lee et al., 2016) dataset is a curated subset of the Digital Database for Screening Mammography (DDSM), designed to support computer-aided detection and diagnosis of breast cancer. It consists of digitized film mammography images with annotated regions of interest for two primary lesion types: calcifications and masses. Each case is associated with pathologic diagnosis labels (benign or malignant) and includes detailed metadata such as lesion type, subtlety, and assessment category, with all malignant cases biopsy-proven. The original target labels in CBIS–DDSM consist of three categories: MALIGNANT, BENIGN, and BENIGN_WITHOUT_CALLBACK. For this study, we merged BENIGN_WITHOUT_CALLBACK and BENIGN into a single class, thereby formulating a binary classification task. Since the labels are well balanced, we report accuracy rather than ROC AUC for binary classification. Prior work on CBIS–DDSM typically reports high accuracy by augmenting the dataset with additional external sources. In contrast, studies that rely solely on CBIS–DDSM often employ strategies such as resampling the training and test splits to balance class distributions before performing classification. In our experiments, however, we used the dataset in its original form without any modifications.`https://www.cancerimagingarchive.net/collection/cbis-ddsm/`, `https://www.kaggle.com/datasets/awsaf49/cbis-ddsm-breast-cancer-image-dataset` (The Kaggle dataset contains the same images, but stored in resized JPEG format with smaller dimensions to reduce the overall dataset size.)

**PetFinder Adoption**    The PetFinder Adoption(Kaggle, 2019) dataset, released as part of the Kaggle PetFinder.my adoption prediction challenge, contains data for over 15,000 pet profiles aimed at predicting adoption speed. Each sample includes a photo of the pet, descriptive text, and structured tabular attributes such as age, breed, gender, vaccination status, and sterilization status. The adoption outcome is categorized into five classes indicating the time until adoption, with the data exhibiting significant class imbalance. The images are provided in JPEG format and collected from various user submissions, thereby resulting in different image sizes that require preprocessing. `https://www.kaggle.com/competitions/petfinder-adoption-prediction`

**Salary**    The Salary dataset consists of job postings in India with the task of predicting salaries. It includes 19,802 training and 6,601 test samples. Each entry contains company (encoded), years of experience, job description, designation, job type, key skills, and location, with salary range as the prediction target. `https://www.kaggle.com/datasets/ankitkalauni/predict-the-data-scientists-salary-in-india`

**Women's Clothing E-Commerce**    The Cloth dataset consists of 23,486 customer reviews with 10 feature variables. Each sample contains textual attributes (title, review body) and structured metadata (Clothing ID, age, rating, recommendation indicator, positive feedback count, division, department, class). Brand references were anonymized by replacing company names with "retailer." The dataset offers a multimodal setting for tasks such as sentiment analysis, recommendation prediction, and text–metadata interaction modeling. `https://www.kaggle.com/datasets/nicapotato/womens-ecommerce-clothing-reviews`

## A.4 EXPERIMENTS ABOUT ACTIVATION ALGORITHM IN FEATURE GENERATION LAYER

Since GLU reduces the dimensionality by half after activation, the first and second affine layers differ in size. To ensure a fair comparison independent of parameter count—as performance often improves with an increased number of parameters—we maintained the first layer's dimension unchanged under GELU and projected to the PFT input dimension (192) only in the second layer. This configuration results in a GELU-based network with a larger parameter budget.

Table 7: Comparison of accuracy and mean output-vector orthogonality when using GELU versus GLU as the activation function in MGM.

| Activation | CBIS–DDSM(MASS) | | Salary | |
|---|---|---|---|---|
| | Performance | Orthogonality | Performance | Orthogonality |
| GELU | 72.09 | 0.0565 | 45.04 | 0.04876 |
| GELU(w/ Ortho loss) | 72.23 | 0.0874 | 45.28 | 0.06237 |
| GLU | 75.10 | 0.0913 | 45.87 | 0.05831 |
| GLU(w/ Ortho loss) | 73.12 | 0.1247 | 44.95 | 0.06728 |

## A.5 ENCOURAGING DIVERSITY IN FEATURE EMBEDDINGS

We further examined whether MGM can generate diverse and independent features from unstructured data through its gating mechanism, and how this diversity influences downstream performance. As a proxy for independence, we measured the orthogonality of embedding vectors. Table 7 reports results on the CBIS–DDSM (MASS) dataset and the Salary dataset. We compared a baseline multi-head MLP encoder with GELU activation against MGM, which uses a GLU-based multi-head MLP. Across both datasets, MGM consistently produced embeddings with higher orthogonality than the baseline. To assess the role of orthogonality more directly, we augmented the baseline encoder with an auxiliary loss term defined as the inverse of pairwise orthogonality, encouraging more diverse embeddings. This modification yielded only a small, within-error-margin improvement over the baseline and did not enhance MGM's performance. These findings suggest that while explicit orthogonality regularization can promote diversity, it may also introduce noise that undermines representational quality. Overall, adaptive gating in MGM provides a more effective and robust mechanism for achieving useful feature diversity than direct orthogonality constraints.

## A.6 DETAILS OF COSINE SIMILARITY COMPUTATION FOR MULTIMODAL FEATURE EMBEDDINGS

We present the detailed implementation of the cosine similarity–based correlation analysis described in Figure 3 and Subsection 5.4. The TabPFN encoder for tabular data groups multiple features into a single embedding to reduce memory usage. However, when comparing cosine similarity between tabular and image (or text) embeddings, such grouping may introduce noise. To obtain more accurate relationships, we set the group size of tabular features to 1, generating individual embeddings for each feature and then comparing them with image embeddings.

Cosine similarity between input features was computed at the instance level and then averaged across instances. While averaging over the entire dataset before computing similarity vectors reduces computational cost, it can underrepresent the influence of individual samples. On the other hand, computing similarity for every token embedding is computationally expensive and makes it difficult to visualize overall relationships. Therefore, we adopted an intermediate compromise, as described in Algorithm X below.

The relatively low cosine similarity among tabular features in Figure 3 can be attributed to the high-frequency variation inherent in tabular data(Beyazit et al., 2023). Such rapid fluctuations across samples prevent the feature vectors from clustering around a single dominant direction in the embedding space. Instead, they disperse over a broader angular range. Since cosine similarity primarily measures directional alignment rather than magnitude, this dispersion naturally yields lower average similarity values, reflecting the reduced coherence in feature orientations.

### A.7 ROBUSTNESS OF PFN IN THE LOW-DATA REGIME

The robustness of our MMPFN model under small training sets stems from the strong priors learned during large-scale meta-training on synthetic datasets(Hollmann et al., 2022; 2025). These priors capture a wide range of plausible tabular distributions, enabling effective generalization even when only limited real samples are available. Fine-tuning adapts the pretrained PFN to task-specific patterns, but the Bayesian inference–like formulation continues to regularize learning by integrating over plausible predictors. Because the model requires only light adaptation rather than full parameter optimization, it avoids overfitting and remains stable in low-sample settings. Together with the inductive bias of the Per-Feature Transformer, this explains the superior performance of our fine-tuned PFN across low-data experiments.

### A.8 REGRESSION TASKS WITH MMPFN

Table 8: Regression experiments were run with five random seeds; results are reported as mean (std). CD18 is evaluated with $R^2$, and Pawpularity with RMSE.

|  | modality | CD18 | Petfinder-Pawpularity |
|---|---|---|---|
| BogOfTricks | T | 0.737 | - |
| CHARMS | T | - | 18.431 |
| TabPFN | T | 0.764(0.016) | 21.490(0.747) |
| MMPFN | T+I | **0.769(0.014)** | 20.121(0.294) |

TabPFN, starting from version 2, introduced support for regression tasks. Since MMPFN is built on TabPFN v2, it can also be applied to regression problems. To evaluate its performance, we conducted experiments on two datasets: **CD18** and **Petfinder Pawpularity**.

The CD18 dataset was originally used in Tang et al. (2024b). The results are summarized in Table 8, where the reported scores are: Bag of Tricks (0.737), TabPFN (0.764), fine-tuned TabPFN (0.764), and MMPFN (0.769). These results indicate an improvement, although most of the gain can be attributed to the strong baseline performance of TabPFN. The additional contribution of image features was marginal. However, this does not imply that MMPFN inherently struggles with regression tasks. As illustrated in Figure X, the degree of improvement depends on whether each modality contributes sufficiently independent information. While the improvement in this case was minor, the observable gain from incorporating image features suggests that MMPFN retains potential for broader regression tasks.

The second dataset, Petfinder Pawpularity, was evaluated differently. In Jiang et al. (2024), results were reported in RMSE, with a score of 18.431. Our MMPFN experiments yielded an RMSE of 20.121, which appears worse by this metric. However, when evaluated in terms of $R^2$, both models achieved values below 0.1, indicating very weak predictive power. This suggests that simply optimizing for lower RMSE does not necessarily reflect a meaningful model fit. Therefore, we exclude the Pawpularity dataset results from our final evaluation.

### A.9 POTENTIAL EXTENSIONS FOR IMPROVING MMPFN PERFORMANCE

Table 9: Performance of MMPFN when replacing the image encoder from DINOv2 to DINOv3. Results are reported as mean (std) over five random seeds.

| image encoder | PAD–UFES–20 | CBIS-DDSM(MASS) | CBIS-DDSM(CALC) | PETFINDER |
|---|---|---|---|---|
| DINOv2 | 84.87(1.72) | 75.56(1.29) | 76.07(0.85) | 40.74(1.36) |
| DINOv3 | *85.61(0.78)* | 75.48(0.93) | *76.75(1.20)* | 40.57(1.44) |

**Replacing Foundation Models** MMPFN is designed by integrating two pretrained foundation models, which makes it naturally extensible as newer and stronger foundation models become available. This modularity enables straightforward replacement of components such as TabPFN or DINO

with more recent architectures. By leveraging improved encoders, one can expect higher-quality feature representations, thereby enhancing downstream performance. For instance, substituting DI-NOv2 with DINOv3, which was released very recently, yielded measurable gains, as reported in Table 9. On the PAD–UFES–20 dataset, accuracy increased by approximately $0.74$ percentage points, demonstrating the benefit of adopting more advanced pretrained encoders. Likewise, using TabPFNv2(Hollmann et al., 2025) consistently outperformed TabPFNv1(Hollmann et al., 2022), and MMPFN adopts TabPFNv2 as the default encoder in all experiments.

**Fine-tuning Encoders**   Many prior studies adopt frozen foundation models, fine-tuning only the task-specific head, especially in small-scale or resource-constrained settings. Nonetheless, full or partial fine-tuning is also widely employed, depending on dataset size, computational budget, and research objectives (Kumar et al., 2022; Lin et al., 2022). In our experiments, given the relatively limited scale of datasets, we followed the frozen-encoder setting and fine-tuned only the subsequent layers. However, in larger-scale or higher-resource scenarios, fine-tuning the foundation model itself could further improve performance, offering an additional avenue for extending MMPFN.

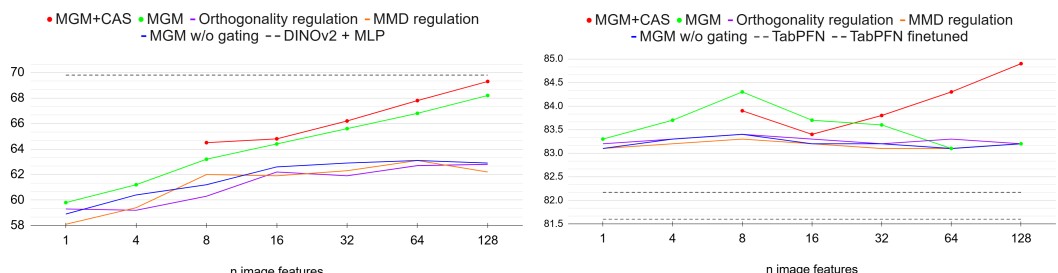

Figure 7: Under the same experimental conditions as Figure 2, we additionally report the performance of models incorporating Orthogonality and MMD as regularization constraints.

### A.10 Evaluating Distribution Alignment Across Modalities

We investigated whether applying embedding space alignment techniques—commonly used in multimodal learning—could improve the performance of MMPFN. Specifically, we incorporated Maximum Mean Discrepancy (MMD) (Gretton et al., 2012), which measures distributional differences(Zhu, 2024), into our modeling framework. Since MMD can quantify discrepancies between embedding distributions, we applied it to embeddings generated by each feature. For tabular data, where feature distributions vary substantially, we employed Joint MMD (JMMD) (Long et al., 2017) to measure distributional differences across the joint distribution of features.

We trained the multi-head MLP module that generates unstructured feature embeddings by including the computed discrepancy between tabular embeddings and image/text embeddings as an additional loss term. However, as shown in Figure 7, this approach consistently underperformed compared to MGM. Incorporating this alignment loss into MGM also failed to improve performance.

Cosine similarity analysis and these negative results suggest that feature embeddings extracted by MGM from image and text modalities are already sufficiently mapped into a semantically compatible space with tabular embeddings, enabling effective interaction in the attention module. Moreover, while methods such as MMD can reduce distributional gaps, they may inadvertently remove discriminative information, leading to performance degradation. We therefore infer that when embeddings from MGM and CAS are already aligned into a sufficiently similar space, enforcing explicit distributional alignment does not necessarily yield further gains.

### A.11 Adjustment of the features_per_group Parameter.

MMPFN embeds tabular features in the same way as TabPFN, allowing them to be grouped. Each group can contain either one or two features, with the default set to two. Setting the number of features_per_group to 1 increases memory consumption, which limits the number of tabular features that can be used as input. However, this configuration can yield improved performance, especially when there is a small number of tabular features. We attribute this effect to two main

factors. First, since each tabular feature is embedded individually, the model can capture feature-specific variations more directly, leading to richer representations. Second, increasing the number of tabular feature tokens allows us to proportionally increase the number of multimodal input tokens, maintaining a better token balance across modalities. This, in turn, enhances the amount of information extractable from different modalities.

For example, in the *Salary* dataset, only four tabular features (categorical and numerical, excluding text attributes) are available. With the default setting of TabPFN, these four features are divided into two groups, resulting in only two tabular embeddings. Although it is reasonable for text embeddings to outnumber tabular embeddings given the richer information in text, excessive token imbalance risks underrepresenting the tabular modality. By setting `features_per_group` = 1, we increase the number of tabular embeddings to four, thereby allowing us to also increase the number of text embeddings without introducing severe imbalance. This configuration yields higher performance compared to `features_per_group` = 2. Nonetheless, one must note that this approach is constrained by memory usage.

### A.12 TRANSFORMER TOKEN OUTPUTS FOR FEATURE GENERATION

Transformer-based language encoders (Clark et al., 2020) and image encoders based on Vision Transformers (ViTs) (Oquab et al., 2023) provide output embeddings for each individual token. While one could consider using all token outputs instead of the aggregated $[CLS]$ token for feature generation, this approach significantly increases memory overhead. For instance, when resizing PAD–UFES–20 images to $336 = 14 \times 24$ pixels and encoding them with the DINOv2 ViT-B/14 backbone, the model produces 576 token embeddings—more than four times the number of MGM heads (128) used in our experiments. In terms of storage, the $[CLS]$ embedding requires only 7.1 MB, whereas the full token outputs occupy 4.1 GB, leading to a substantial increase in memory usage. A similar issue arises with text data: in the Cloth dataset, most text attributes approach the maximum input length of 512 tokens, which likewise results in prohibitive memory consumption when retaining all token embeddings. Moreover, our experiments demonstrate that models leveraging token outputs perform worse than those relying on the $[CLS]$ embedding, with accuracy decreasing by 0.85 percentage points (84.02% vs. 84.87%). We attribute this to the fact that the $[CLS]$ representation of a well-trained foundation model encodes task-relevant global information, whereas raw token outputs often contain redundant or noisy features. Consequently, they are less suitable for transformation into tabular-like feature representations.

### A.13 MULTI-HEAD GATED MLP

---

**Algorithm 1:** Multihead GLU Image Projector

---
**Input:** Input $x$, $n\_heads$, $in\_dim$, $h\_dim$, $out\_dim$
**Output:** Projected tensor
projs $\leftarrow$ [];
**for** $i = 1$ **to** $n\_heads$ **do**
    Define proj$_i$ as:;
        LayerNorm($in\_dim$);
        Linear($in\_dim$, $2h\_dim$);
        GLU;
        Dropout(0.1);
        Linear($h\_dim$, $out\_dim$);
    Append proj$_i$ to projs;
**end**
$outs \leftarrow$ [proj($x$) for proj in projs];
**return** Concatenate($outs$, dim$= -2$)

---

### A.14 LLM USAGE

In this study, we employed LLM(ChatGPT5) for two purposes. First, ChatGPT5 is used to improve the fluency of English text by evaluating and correcting grammar and word choice. Second,

ChatGPT5 was used to summarize cited papers for quick content review or to retrieve specific information. LLMs did not intervene in the ideation process or in model development.

