# OpenReview forum: "MultiModalPFN: Extending Prior-Data Fitted Networks for Multimodal Tabular Learning"
_ICLR.cc/2026/Conference — ICLR 2026 Conference Withdrawn Submission_

### Official Review · Reviewer_JCqE · 2025-10-20

**Soundness:** 2
**Presentation:** 1
**Contribution:** 2
**Rating:** 2
**Confidence:** 4

**Summary:**

This paper proposes MMPFN, a multimodal prior-data fitted network that handles tabular and non-tabular modalities in a unified manners to address modality heterogeneity. This model includes a multi-head gated MLP (MGM) that aims to address overcompressed non-tabular embeddings and a cross-attention sampler (CAS) that aims to mitigate the attention imbalance issue.

The proposed  model is evaluated on image-tabular and text-tabular datasets and compared against prior multimodal approaches.

**Strengths:**

* S1: This paper explores image-tabular representation learning, which is of interest to the community.

* S2: The insight of attention imbalance and the experiment of using different number of image features is interesting.

**Weaknesses:**

**Major Weakness**

* W1: Methodology details for MGM and CAS are not efficient. Providing some semantic illustrations or necessary equations could help readers to understand.

* W2: The novelty of this paper appear to be limited. The proposed method seems to be a straightforward adaptation of PFN to the multimodal domain. Gated MLP and cross-attention sampler have been studied in prior works [R1].

* W3:  The attention imbalance analysis is superficial. Assuming similar expected attention weights across modalities is overly simplistic. actually they can very different (Figure 3 shown in the paper). Furthermore, Figure 3 suggests notable differences. Furthermore, Equation 3 implies balancing token numbers could mitigate imbalance, yet Figure 2 shows the best results with 8 image features, this inconsistency is not yet discussed.

* W4: To address token over-compression, it is unclear why the authors use gated MLPs to expand the [CLS] token instead of leveraging non-[CLS] tokens in the embedding sequence.

* W5: The experimental settings of comparing methods are unclear. TIP and MMCL use ResNet-50 as the image encoder but MMPFN uses ViT-B. In addition, TIP and MMCL are SSL pre-training strategies. Do the authors conduct pre-training from TIP and MMCL before comparing with MMPFN? It is doubtable whether the performance gain comes from a unfair comparison.

* W6: The paper’s presentation needs refinement. Structural organization, clarity, and proofreading (e.g., “while mitigating” in line 21) all require attention.

[R1] Li, Junnan, et al. "Blip-2: Bootstrapping language-image pre-training with frozen image encoders and large language models." ICML 2023.

**Questions:**

**Primary  Questions/Suggestions**

* QS1: What distinguishes MMPFN from prior multimodal PFN adaptations? Please clarify the methodological novelty.

* QS2: Could the authors provide more detailed descriptions (and possibly equations) for MGM and CAS?

* QS3: How do you ensure fairness in comparisons?

* QS4: Improving the presentation of the paper could make it better

---

### Official Review · Reviewer_YaBg · 2025-10-31

**Soundness:** 2
**Presentation:** 3
**Contribution:** 2
**Rating:** 4
**Confidence:** 3

**Summary:**

This paper proposes MMPFN (multimoda prior data fitted network) that extends TabPFN to handle multimodal inputs (e.g., tabular + image). The authors basically use modality-specific encoders and project non-tabular embeddings to tabular embedding space and feed all the featuers to TabPFN backbone for inference. The projector relies on multi-head gated MLP (expands cls token into multiple tokens) and cross-attention sampler to avoid token-count induced attention imbalance. The authors showed on a few medical and general multimodal benchmarks that their proposed architecure outperforms baselines while requiring a lot less finetuning compute.

**Strengths:**

- the motivation for the proposed designs seems to be clear
- the evaluation is pretty broad, e.g., covering medical and general datasets and the performance gain seems to be consistently better.
- the paper is relatively simple to read/parse!

**Weaknesses:**

- the main issue with the paper is that the proposed architecture changes are pretty ad-hoc. given the specificity of the task (e.g., image/text + tabular dataset), and the added complexity of the proposed architecture, it's arguable whether this contribution has lots of impact or meaningfully progress even this specific direction of improved multi-modal tabular model.
- the main result of performance comparison to baselines in Table 1 and 2 can be strengthened if the authors can be more explicit about the encoders / components used in these baselines and/or dataset used for pretraining/finetuning. Because without apple-to-apple comparison, we are not sure if the gain is from the dataset, the modeling, or the scale of the model, etc.
- Comparing TabPFN vs. MMPFN, it seems that the performance gain from (1) architecture improvement (2) relying on image information as well gives pretty modest gains. It would be nice to know relative proportion of contribution to performance gain from architecture change vs. adding image modality.
- can author explain the reasoning that finetuning is done with batch size of 1 with max training step of 100? does it mean the mdoel is finetuned on 100 examples only. It would be useful/informative if the authors could provide some reasoning on this choice and some ablations on varying batch size and fintuning step's affect on performance so that we have a better idea the sensitivity of the proposed approach to these very important hyperparameters.

**Questions:**

n/a

---

### Official Review · Reviewer_vrdT · 2025-11-01

**Soundness:** 3
**Presentation:** 3
**Contribution:** 3
**Rating:** 4
**Confidence:** 3

**Summary:**

This paper presents Multi-Modal Prior-data Fitted Network (MMPFN), a framework that extends the TabPFN model to handle datasets containing a mix of tabular, image, and text data.
The architecture uses separate pre-trained encoders for each modality and uses a trainable modality projector to transform the text and image features. In this way, the newly added image and text can be integrated into the whole model for downstream tasks.

**Strengths:**

The motivation of this paper is clear: adding new modalities to tabular networks.

This paper is well-written and easy to follow.

**Weaknesses:**

This paper chooses a Q-former (Li et al. 2023) projector to convert the image and test features. Why choose the Q-former to transform vision features with Llava (Liu et al. 2023) have demonstrated that MLP has better convergence and data efficiency than Q-former.  I think the justification for this design is not sufficient.

The performance of the proposed method is not sufficient; it would be better to show other advantages of this method against the baselines.  On some sub-tasks, the old baselines showcase an obvious performance advantage, and the proposed method is a data-friendly framework, so why does the performance lag behind the baselines?

The technical contribution of this paper is not significant the major module shares a similar design with Q-former.

**Questions:**

Will finetuning TabPFN cause the performance drop on single modality tasks? Can lora obtain better performance?

---

### Note · Authors · 2025-11-12

**Comment:**

We respectfully request to withdraw our submission from consideration. Upon further evaluation, we identified aspects of the work that require revision to ensure the quality of our result. We appreciate your understanding and thank the reviewers and area chairs for their time and effort.

**Withdrawal Confirmation:**

I have read and agree with the venue's withdrawal policy on behalf of myself and my co-authors.